# Fe_3_N Nanoparticle-Encapsulated N-Doped Carbon Nanotubes on Biomass-Derived Carbon Cloth as Self-Standing Electrocatalyst for Oxygen Reduction Reaction

**DOI:** 10.3390/nano13172439

**Published:** 2023-08-28

**Authors:** Yongxin Zhao, Dandan Liu, Yubin Tian, Yuzhu Zhai, Chaofan Tian, Sen Li, Tao Xing, Zhi Li, Pengcheng Dai

**Affiliations:** 1College of New Energy, China University of Petroleum (East China), Qingdao 266580, China; 2College of Textile and Clothing, State Key Laboratory of Bio-Fibers and Eco-Textiles, Collaborative Innovation Center for Eco-Textiles of Shandong Province, Qingdao University, Qingdao 266101, China; 3New Energy Division, National Engineering Research Center of Coal Gasification and Coal-Based Advanced Materials, Shandong Energy Group Co., Ltd., Jining 273500, China; 4School of Materials Science and Engineering, Xi’an Jiaotong University, Xi’an 710049, China

**Keywords:** Fe_3_N nanoparticles, carbon nanotubes, biomass, oxygen reduction reaction, Zn−air battery

## Abstract

The design and fabrication of low-cost catalysts for highly efficient oxygen reduction are of paramount importance for various renewable energy-related technologies, such as fuel cells and metal–air batteries. Herein, we report the synthesis of Fe_3_N nanoparticle-encapsulated N-doped carbon nanotubes on the surface of a flexible biomass-derived carbon cloth (Fe_3_N@CNTs/CC) via a simple one-step carbonization process. Taking advantage of its unique structure, Fe_3_N@CNTs/CC was employed as a self-standing electrocatalyst for oxygen reduction reaction (ORR) and possessed high activity as well as excellent long-term stability and methanol resistance in alkaline media. Remarkably, Fe_3_N@CNT/CC can directly play the role of both a gas diffusion layer and an electrocatalytic cathode in a zinc–air battery without additional means of catalyst loading, and it displays higher open-circuit voltage, power density, and specific capacity in comparison with a commercial Pt/C catalyst. This work is anticipated to inspire the design of cost-effective, easily prepared, and high-performance air electrodes for advanced electrochemical applications.

## 1. Introduction

The electrochemical oxygen reduction reaction (ORR) is a critical process in air electrodes in fuel cells and metal–air batteries [1,2]. It determines the overall chemical-electrical conversion efficiency of these electrochemical devices due to the complicated 4e^−^ reaction pathways, sluggish reaction kinetics, and high overpotential [3]. Platinum (Pt)-based composite materials, due to their high catalytic activity, are commonly recognized as state-of-the-art ORR electrocatalysts [3,4]. However, their large-scale applications have been severely hindered by serious drawbacks, such as scarce reserves, prohibitive costs, insufficient long-term durability, and vulnerability to methanol crossover [5,6]. Consequently, extensive efforts have been devoted to developing cost-effective electrocatalysts with competitive activity and excellent stability as alternatives to Pt-based materials.

Iron nitrides with rich N content in a carbon matrix (Fe_x_N-C) containing active Fe-N-C sites [7], as a kind of newly developed non-noble metal electrocatalysts [8], present enhanced catalytic activity [9], low cost, and strong methanol tolerance [10], and have been recognized as promising alternatives to Pt-based catalysts [11,12]. For instance, Park et al. prepared a novel Fe_3_N/C nanocomposite material for ORR via pyrolysis of carbon black with iron-containing precursors under an NH_3_ atmosphere. The material consists of highly dispersed iron nitride nanoparticles loaded on nitrogen-doped carbon and exhibits excellent ORR activity and direct four-electron pathway in alkaline solutions [13]. However, Fe_x_N nanoparticles are thermodynamically unstable and prone to migration and coalescence during the catalysis process because of their high surface energy, resulting in an apparent decrease in activity and stability [14]. Encapsulating Fe_x_N nanoparticles in the nanoshells or nanopores of a carbon matrix is one of the most promising strategies to overcome the stability issue and benefit the generation of Fe-N-C active sites [15]. Zheng et al. prepared a graphene framework with nitrogen-doped carbon nanotubes with encapsulated Fe/Fe_3_N nanoparticles through a one-step calcination strategy, which has high stability and high discharge battery voltage, close to platinum/carbon in zinc–air batteries [16]. However, most of the reported preparation strategies of carbon-encapsulated Fe_x_N nanoparticles involve complex procedures, such as the pre-synthesized pyrolysis with metal–organic frameworks as precursors [3] and the polymer carbonization–acid etching route [17]. Some synthetic methods employ expensive iron and carbon precursors, such as iron phthalocyanine [18] and porphyrin [19], which increase the cost and pose challenges for large-scale synthesis. Moreover, Fe_x_N-C catalysts are generally connected to a gas diffusion layer to fabricate a gas electrode with ionomers such as Nafion. Li et al. recently employed cesium salt of phosphotungstic acid as a proton conductor to replace Nafion ionomers in MEAs and explore its degradation mechanism and found that Nafion ionomers can be easily deactivated by the generated ·OH radicals, leading to fast performance degradation [20]. By comprehensively evaluating costs, activity, and stability, the direct growth strategy of Fe_x_N nanoparticles encapsulated in carbon nanomaterials on gas diffusion electrodes (i.e., carbon cloth) is highly desirable, yet few researchers have focused on this direction.

In this study, we report that Fe_3_N nanoparticle-encapsulated N-doped carbon nanotubes grown on the surface of a flexible carbon cloth (denoted as Fe_3_N@CNTs/CC) can be synthesized via simple one-step calcination of cotton cloth, one of the most commonly used biomass products. Fe_3_N nanoparticles were produced during the carbonization process of the cotton cloth under an ammonia (NH_3_) atmosphere and catalyzed the growth of N-doped carbon nanotubes on the generated carbon cloth, resulting in the overall fabrication of a Fe_3_N@CNTs/CC hierarchical structure. This hierarchical structure is capable of enlarging the surface area, promotes the exposure of active species, and gives rise to a remarkable ORR activity with a +50 mV higher half-wave potential (E_1/2_ = 0.91 V) than Pt/C (E_1/2_ = 0.86 V) in alkaline media. The long-term stability and methanol tolerance of Fe_3_N@CNTs/CC are also significantly improved compared to Pt/C. Meanwhile, as a flexible and free-standing electrocatalyst, Fe_3_N@CNTs/CC can be used directly as the air cathode in Zn–air batteries (ZIBs) and exhibits higher open-circuit voltage (1.50 V vs. 1.42 V), power density (157 mW cm^−2^ vs. 124.5 mW cm^−2^), and specific capacity (814.9 mAh g^−1^ vs. 743.9 mAh g^−1^) than the Pt/C based gas electrode.

## 2. Materials and Methods

### 2.1. Materials and Chemicals

Ferric chloride hexahydrate (analytical pure grade), zinc acetate dihydrate ≥ 98%, potassium hydroxide (analytical pure grade), concentrated sulfuric acid 98%, and ethanol 99% were purchased from Sinopsin Group Chemical Reagent Co., Ltd. (Shanghai, China) Commercial Pt/C catalyst (20 wt%) was purchased from Sigma-Aldrich (Shanghai, China), Nafion solution (5%) and Teflon dispersion (60%) were obtained from Dupont (Wilmington, DE, USA), Vulcan XC 72R (99.9%) was obtained from Johnson Matthey (London, UK), and cotton cloth (α-cellulose > 90%) was purchased from Aubang Technology Co., Ltd. (Foshan, China), Deionized water was made in the laboratory.

### 2.2. Preparation of Fe_3_N@CNT/CC

The cellulose cotton cloth (5 g) was washed with deionized water thrice and dried in an oven at 80 °C. Next, an impregnation solution was prepared by dissolving 2.7 g of FeCl_3_·6H_2_O in 5 mL of deionized water at 60 °C. The impregnation solution was then dropped evenly onto the cellulose cotton cloth and dried at 80 °C. The impregnated cotton cloth was then annealed in a tube furnace at 1000 °C for 1 h under an NH_3_ atmosphere.

### 2.3. Physicochemical Characterization

Material characterization: An X-ray diffractometer (XRD, χPert Pro, Panaco, The Netherlands) was used to characterize the phase composition of the samples. A field-emission scanning electron microscope (SEM, JSM-7500F, JEOL, Musashino, Tokyo) and a transmission electron microscope (TEM, JEM 2100F, JEOL, Musashino, Tokyo) were used to observe the surface morphology of the materials, and energy-dispersive X-ray spectroscopy (EDS) measurements were performed using an transmission electron microscope (Tecnai G20, FEI, OR, USA). X-ray photoelectron spectroscopy (XPS) was used to characterize the materials’ composition, element content, and valence state. The pore structure and specific surface area of the materials were characterized by N_2_ adsorption−desorption isotherms (Autosorb-iQ2, Quantachrome, FL, USA) at 77 K. The pore-size distribution plot was measured using the Barrett–Joyner–Halenda (BJH) model.

### 2.4. Electrochemical Measurements

Electrochemical measurements were conducted using a typical three-electrode system of a CHI760E electrochemical workstation with 0.1 M KOH as the electrolyte at 25 °C. The platinum wire and Ag/AgCl electrodes were used as the opposite and reference electrodes, respectively. The catalyst-modified glassy carbon rotating disk electrode (RDE) was used as the working electrode (0.1256 cm^2^). To prepare the catalyst ink, 4 mg of catalyst was mixed with 500 µL of complex solvent (the volume ratio of ethanol/deionized water/Nafion = 9:36:5) and ultrasound for 30 min. The prepared 10 µL catalyst ink droplet was deposited on the turntable electrode (RDE) with a catalyst load of 0.31 mg cm^−2^.

During the test, cyclic voltammetry (CV) was tested in 0.1 M KOH solution saturated with O_2_ or N_2_ at a scan rate of 10 mV s^–1^. Linear sweep voltammetry (LSV) was tested in 0.1 M KOH solution saturated with O_2_ or N_2_ at a scan rate of 5 mV s^–1^.

The ORR kinetic currents and the number of transferred electrons were calculated using the K-L equations:j^−1^ = j_k_^−1^ + (j_d_)^−1^ = j_k_^−1^ + (Bω^0.5^)^−1^
(1)
B = 0.2nFC_0_D_0_^2/3^ν^−1/6^
(2)
where j is the measured current density (j), j_K_ is the kinetic current, j_d_ is the diffusion-limiting current, ω is the rotating speed (revolutions per minute), n is the number of transferred electrons, F is the Faraday constant (96,485 C mol^−1^), C_0_ is the bulk concentration of O_2_ (1.2 × 10^−6^ mol cm^−3^), D_0_ is the diffusion coefficient of O_2_ (1.9 × 10^−5^ cm^2^ s^−1^), and ν is the kinetic viscosity (0.01 cm^2^ s^−1^).

In the rotating ring-disk electrode (RRDE) test, the measured values of the ring and disk currents were used to calculate the H_2_O_2_ production and the number of electron transfers (n) using the following equations:Y_H2O2_ = 2 × (I_r_/N)/(I_d_ + I_r_/N) × 100% (3)
n = 4I_d_/(I_d_ + I_r_/N) (4)
where I_d_ denotes the disk current; I_r_ represents the ring current; and N is the current collection efficiency of the Pt ring with a value of 0.36 from the reduction of K_3_[Fe(CN)_6_]. LSV measurements via the RRDE tests were carried out at a rotating rate of 1600 rpm and a scan speed of 5 mV s^−1^. Chronoamperometry was conducted at 0.6 V (vs. RHE) for 10,000 s (1600 rpm) in O_2_-saturated 0.1 M KOH to investigate the durability of the catalyst [21].

### 2.5. Zn–Air Battery Tests

A Zn sheet that had been polished with sandpaper was used as the anode. Fe_3_N@CNT/CC was applied as the air cathode. The aqueous electrolyte comprised 6 M KOH and 0.2 M zinc acetate. For comparison, Pt/C-based air electrode was prepared as follows: 1 mg of Pt/C catalyst, 400 μL of anhydrous ethanol, and 10 μL of Nafion were mixed evenly and then deposited on the commercial carbon cloth with a Pt/C loading mass of 0.82 mg cm^−2^.

## 3. Results and Discussion

### 3.1. Structural and Compositional Analyses

Fe_3_N@CNT/CC was prepared via a simple thermal treatment of FeCl3-decorated cellulose cotton cloth under an NH_3_ atmosphere, as illustrated in Figure 1a. The digital images of the Fe_3_N@CNT/CC film in Figure 1b reveal its excellent flexibility, as it can be bent and rolled without structural deterioration. ICP and elemental analysis were performed to identify the composition ratios of the Fe_3_N@CNTs/CC film. As shown in Appendix A, the weight fraction of Fe, C, N, and O is 20.096%, 72.23%, 5.067%, and 2.182%, respectively. Thus, the weight fraction of the ORR active Fe_3_N species was determined to be 21.8%. SEM was used to analyze the morphology of Fe_3_N@CNT/CC. It is observed that the fiber-braided structure of the cotton cloth is well retained during the carbonization process (Figure 1c). Carbon nanotubes are generated and connected to the fibers (Figure 1d) with a diameter of ~100 nm (Figure 1e). Shiny nanoparticles are observed to be encapsulated within the carbon nanotubes (Figure 1e). As shown in the TEM and STEM images (Figure 1f and Appendix A), the diameters of Fe_3_N nanoparticles are in the range of 30–80 nm. The lattice spacing of the nanoparticles is 0.206 nm (Figure 1g), corresponding to the (-1-11) crystal plane of Fe_3_N [21,22]. Additionally, clear lattice fringes are observed outside the nanoparticles with a lattice spacing of 0.34 nm, corresponding to the (002) crystal plane of graphite carbon. The elemental mapping reveals that C and N are evenly distributed throughout the carbon nanotubes, forming N-doped carbon nanotubes. In contrast, Fe is only distributed in the cores of the carbon nanotubes (Figure 1h), indicating Fe_3_N nanoparticles are encapsulated in the carbon nanotubes. In this way, Fe_3_N nanoparticles are trapped in the nanocavity of carbon nanotubes, preventing catalyst agglomeration [23].

X-ray diffraction (XRD) spectrum was recorded to analyze the crystallinity of the obtained Fe_3_N@CNT/CC film (Figure 2a). The diffraction peaks at 38.2°, 41.1°, 43.6°, 57.3°, 69°, and 76.5° correspond to the (110), (002), (-1-11), (-1-12), (300), and (-1-13) crystal planes of Fe_3_N, respectively [3]. The diffraction peak at 26.4° corresponds to the (002) facet of graphite carbon [3], consistent with the HRTEM results. The average crystallite size calculated based on the Scherrer formula is 52 nm, consistent with the STEM result in Appendix A. Figure 2b depicts the Raman spectrogram analysis of the Fe_3_N@CNT/CC film, which exhibits two distinct peaks at 1344 cm^−1^ and 1574 cm^−1^, corresponding to the typical D and G bands of carbon materials [24]. The ratio of I_D_/I_G_ indicates the degree of defect of the carbon material [25]. The I_D_/I_G_ value of Fe_3_N@CNT/CC was determined to be about 0.89, which suggests that less defective or amorphous carbon exists in the as-prepared product. The nitrogen adsorption and desorption isotherm reveal that Fe_3_N@CNT/CC possesses a specific surface area of 337.7 m^2^·g^−1^ (Figure 2c). The pore volume is mainly contributed by micropores, while a certain amount of mesopores also exist with an average size of 4 nm (Figure 2d). The high surface area and pore volume can make the catalytically active sites accessible and ensure fast mass transport efficiency, which enables the catalyst to have significant advantages during electrocatalytic reactions [26,27].

XPS characterizations were further performed to investigate the elements’ specific contents (Appendix A) and chemical states on the surface of Fe_3_N@CNT/CC. The XPS survey spectrum (Figure 3a) reveals the presence of C, N, Fe, and O. The molar fraction of Fe, C, N, and O determined by XPS is 47.1%, 17.9%, 23.7%, and 11.3%. Compared to the ICP and elemental analysis results, the contents of Fe, N, and O determined by XPS are much higher, and the carbon content is lower. This result further proves that Fe_3_N nanoparticles and N-doped carbon nanotubes are grown on the surface of the carbon cloth. In the C 1s spectrum (Figure 3b), the peaks at 284.8 eV, 285.5 eV, and 287.8 eV correspond to C-C, C-N, and C-O signals, respectively [28]. The N 1s (Figure 3c) spectrum is divided into four peaks, including pyridinic nitrogen (398.2 eV), Fe-N_X_ (399.4 eV), pyrrolic nitrogen (401 eV), and graphite nitrogen (402.1 eV) [29]. The ratio of graphitic N, pyrrolic N, pyridinic N, and Fe-N_x_ is 37.6%: 26.3%: 14.6%: 21.5%. The peak corresponding to the Fe-N bond can be observed at 707.8 eV [30]. The Fe 2p spectrum exhibits a peak ascribed to Fe_3_N at the binding energy of 707.8 eV (Figure 3d) [3]. The peaks at the binding energies of 710.7/724.3 and 712.7/726.2 eV originated from Fe^2+^-N/O and Fe^3+^-N/O due to the surface oxidation of the sample [31]. The existence of C-N in the C 1s spectrum, Fe-N_X_ in the N 1s spectrum, and the Fe-N bond in the Fe 2p spectrum verifies the formation of Fe-N-C sites, which have been proven to be efficient ORR active sites [32]. Meanwhile, the graphitized N-doped carbon promotes the electron transfer process, further improving the electrical conductivity of the sample [3].

### 3.2. Electrocatalytic Activities of Fe_3_N@CNT/CC for ORR

The ORR performances of Fe_3_N@CNT/CC were compared with those of commercial Pt/C (Figure 4). Fe_3_N@CNT/CC demonstrated no ORR peaks in the N_2_-saturated electrolyte, while a distinguished reduction peak for ORR appeared when the electrolyte was saturated with O_2_ (Figure 4a). The peak potentials corresponding to Fe_3_N@CNT/CC were 0.91 V, which was 60 mV more positive than that of the corresponding 20% Pt/C (0.85 V) (Figure 4a and Appendix A). The LSV curve of Fe_3_N@CNT/CC (Figure 4b) shows an onset potential of 1.0 V, a half-wave potential of 0.91 V, and a limit current density of 5.6 mA·cm^−1^. In contrast, the initial potential (0.9 V), half-wave potential (0.86 V), and limit current (5 mA·cm^−1^) of the Pt/C catalyst were significantly worse than those of Fe_3_N@CNT/CC. The corresponding Tafel slope of Fe_3_N@CNT/CC (Figure 4c) was 78 mV·dec^−1^, which was lower than that of Pt/C (88 mV·dec^−1^), indicating that Fe_3_N@CNT/CC has a faster catalytic kinetic process [3].

The ORR reactions have two types of electron transfer processes: 2e^−^ transfer process and 4e^−^ transfer process. In the 2e^−^ transfer process, oxygen reacts to produce H_2_O_2_ through the catalytic process [33]. The reaction rate of this process is slow, which is not suitable for application in zinc–air batteries [34]. Moreover, the generated H_2_O_2_ can cause damage to the catalyst and battery components [35,36]. The catalytic product of the 4e^−^ transfer process is H_2_O, making it the required catalytic path for the ORR catalytic reaction [37]. The number of transferred electrons (n) in the catalytic reactions was obtained based on the Koutecký–Levich (K-L) calculation of the LSV curves at different speeds (Figure 4d). As shown in Figure 4e, the average number of transferred electrons of Fe_3_N@CNT/CC was determined to be 3.99, which is close to the theoretical value of 4 and comparable to that of Pt/C. In addition, rotating ring-disk electrode (RRDE) measurements were conducted to explore the ORR pathway of Fe_3_N@CNT/CC (Appendix A). The H_2_O_2_ yield of Fe_3_N@CNT/CC was found to be only 0.14%–0.01% from 0.2 V to 0.8 V (Figure 4f), further proving the 4e^−^ transfer process.

Subsequently, the stability of the Fe_3_N@CNT/CC and Pt/C catalysts was assessed using timed amperometry at 0.8 V (vs. RHE). The current retention of Fe_3_N@CNT/CC decreased by 2.75% during the test at 10,000 s, whereas that of the Pt/C catalyst decreased by 16.5% (Figure 4g). It was observed that Fe_3_N@CNT/CC is more stable during the reaction. Figure 4h depicts the durability test of Fe_3_N@CNT/CC after 2000 CV cycle curves. It can be observed that the half-wave potential value of the catalyst only decreased by 20 mV after 2000 accelerated aging experiments, indicating its excellent stability as well. Methanol resistance is another essential index in ORR. It can be seen from Appendix A that when methanol was added to the electrolyte, the current value of Fe_3_N@CNT/CC decreased by 5%, whereas the current density of Pt/C decreased by 35% (Appendix A), which suggests that Fe_3_N@CNT/CC exhibits outstanding resistance to methanol.

### 3.3. Application of Fe_3_N@CNT/CC Catalyst in ZAB

To further evaluate the practical application of Fe_3_N@CNT/CC, it is essential to assemble it as the air cathode in a ZAB, as illustrated in Figure 5a. The open-circuit voltage of the carbon cloth composed of Fe_3_N@CNT/CC is approximately 1.5 V (Figure 5b), which is higher than that of Pt/C (1.43 V). As shown in Figure 5c, Fe_3_N@CNT/CC can supply a higher voltage at the same current density than Pt/C. The peak discharge power density of the ZAB with Fe_3_N@CNT/CC as the air cathode is 157 mW cm^−2^ at the current density of 300 mA cm^−2^ (Figure 5c,d), which is higher than that of the Pt/C catalyst (124.5 mW cm^−2^). Figure 5e illustrates the constant current discharge test of the battery at a current density of 10 mA cm^−2^. The specific capacity of the Fe_3_N@CNT/CC-based ZAB is 814.9 mAh g^−1^, also higher than that of the Pt/C catalyst (737.9 mAh g^−1^). In addition, Figure 5f depicts the cycling stability of the Fe_3_N@CNT/CC-based ZAB. The voltage range does not widen during the 100 h charge–discharge cycle, indicating the catalyst has excellent stability.

## 4. Conclusions

In this work, Fe_3_N nanoparticle-encapsulated N-doped carbon nanotubes grown on the surface of a flexible carbon cloth were synthesized via simple one-step calcination of the cotton cloth. The structural features and ORR performance of the as-prepared catalysts were investigated in detail. The hierarchical structure can stabilize the active sites, promotes the exposure of active species, and possesses satisfactory ORR performance, including excellent half-wave potential, higher long-term stability, and satisfactory resistance to methanol, outperforming the commercial Pt/C catalysts. Benefiting from the unique structure, Fe_3_N@CNTs/CC can be used directly as a free-standing air cathode in ZABs, exhibiting higher open-circuit voltage, power density, and specific capacity than a Pt/C-based gas electrode. This work paves the way for the construction of cost-effective, easily prepared, and high-performance ORR catalysts and ZAB-based energy conversion devices.

## Figures and Tables

**Figure 1 nanomaterials-13-02439-f001:**
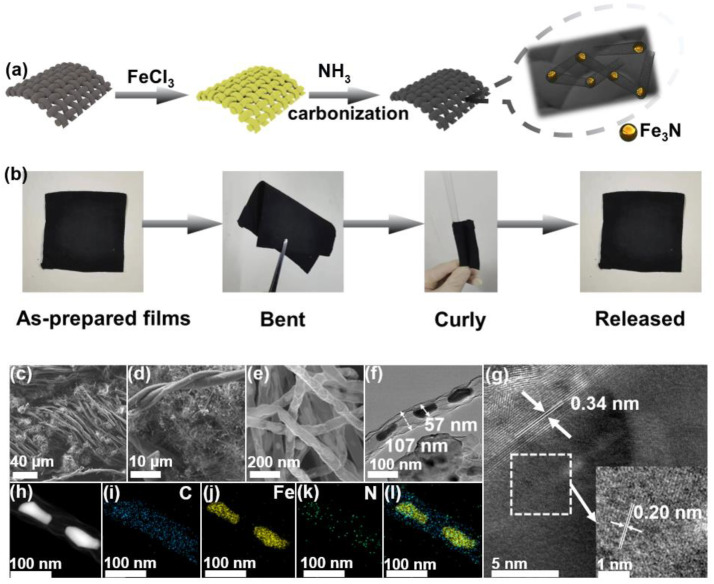
(**a**) Illustration of the synthetic process of Fe_3_N@CNT/CC. (**b**) Digital photos showing the flexibility of Fe_3_N@CNT/CC. (**c**,**d**) SEM images, (**e**) STEM image, (**f**) TEM image, (**g**) HRTEM image, and (**h**–**l**) TEM selective area and its corresponding EDS elemental mappings of Fe_3_N@CNT/CC.

**Figure 2 nanomaterials-13-02439-f002:**
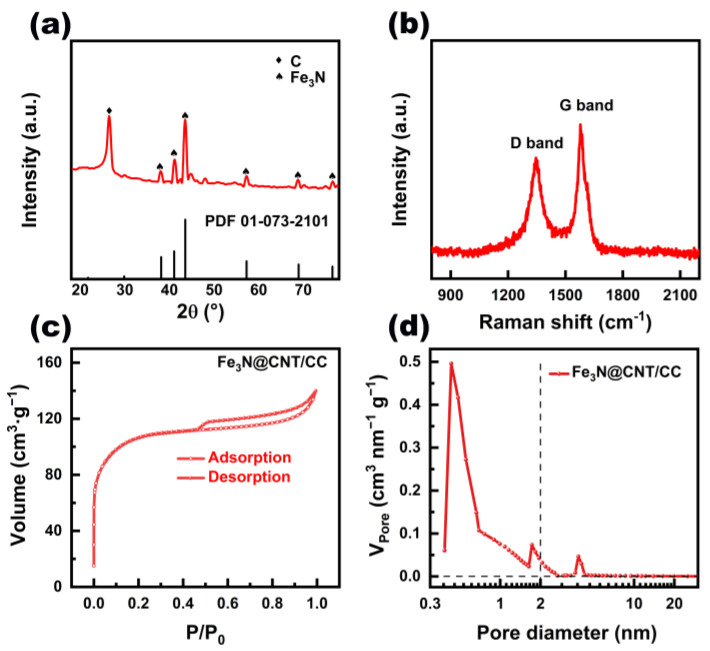
(**a**) XRD pattern, (**b**) Raman spectrum, (**c**) nitrogen adsorption and desorption isotherm, and (**d**) pore-size distribution of Fe_3_N@CNT/CC.

**Figure 3 nanomaterials-13-02439-f003:**
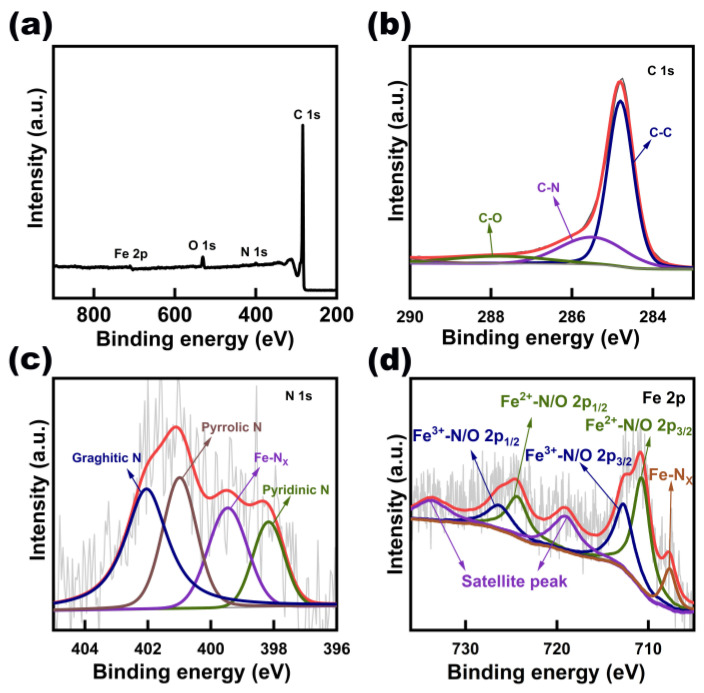
(**a**) XPS survey spectrum, (**b**) high-resolution C 1s, (**c**) high-resolution N 1s, and (**d**) high-resolution Fe 2p spectrum of Fe_3_N@CNT/CC.

**Figure 4 nanomaterials-13-02439-f004:**
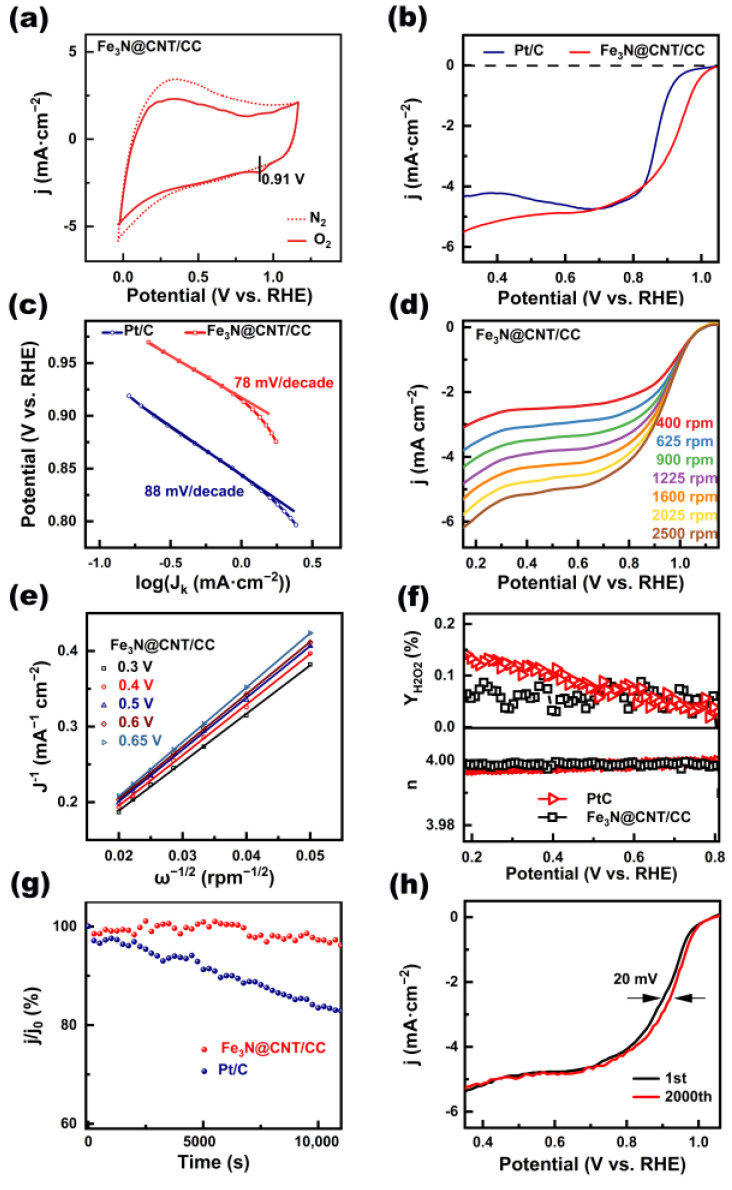
(**a**) CV curves of Fe_3_N@CNT/CC in N_2_ (dotted line)- and O_2_ (solid line)-saturated 0.1 M KOH electrolytes with a scan rate of 10 mV·s^−1^; (**b**) LSV curves of Fe_3_N@CNT/CC and Pt/C, (**c**) corresponding Tafel plots and slops; (**d**) LSV curves of Fe_3_N@CNT/CC at different speeds; (**e**) K-L plot of Fe_3_N@CNT/CC; (**f**) electron transfer number and H_2_O_2_ yield of Fe_3_N@CNT/CC and Pt/C; (**g**) long-term stability test of Fe_3_N@CNT/CC and Pt/C; and (**h**) durability measurement of Fe_3_N@CNT/CC.

**Figure 5 nanomaterials-13-02439-f005:**
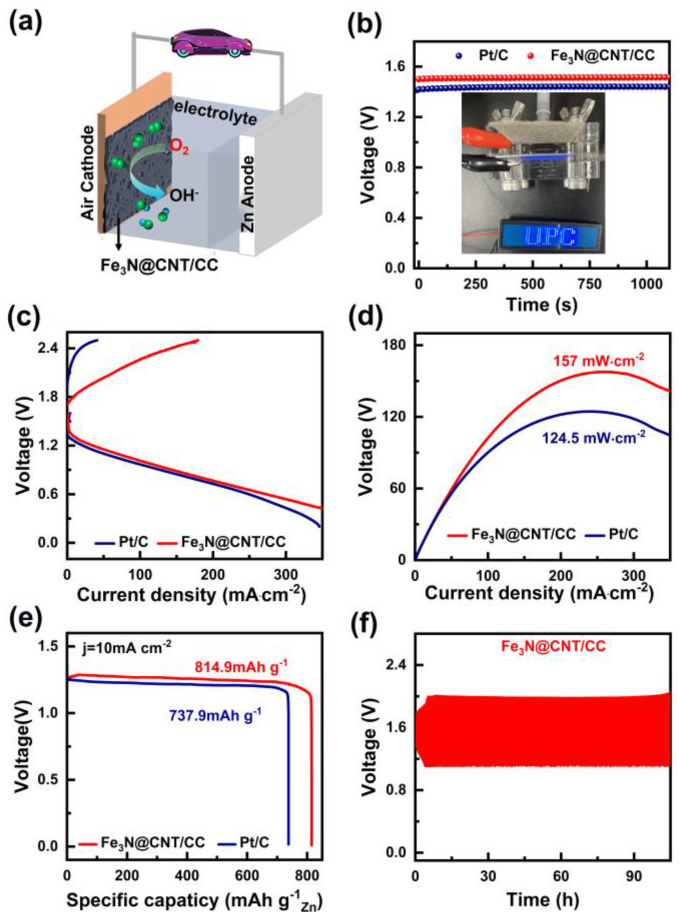
(**a**) Schematic of the as-assembled zinc–air battery, (**b**) open-circuit voltage, (**c**) polarization curves, (**d**) power density curves of the ZABs assembled with Fe_3_N@CNT/CC and Pt/C, (**e**) the discharge specific capacity curves of the ZABs with Fe_3_N@CNT/CC and Pt/C at a current density of 10 mA cm^−2^, and (**f**) the cycling stability of the Fe_3_N@CNT/CC-based ZAB.

## Data Availability

The data presented in this study are available upon request from the corresponding author.

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
