# Peer review of "Fe3N Nanoparticle-Encapsulated N-Doped Carbon Nanotubes on Biomass-Derived Carbon Cloth as Self-Standing Electrocatalyst for Oxygen Reduction Reaction"

_nanomaterials, 2023, doi:10.3390/nano13172439_

Round 1

Reviewer 1 Report

This paper is an interesting report on Fe3N@CNTs/CCs, in which Fe3N particles are loaded on carbon cloth surfaces by ammonia treatment. The Fe3N@CNTs/CCs show higher performance than Pt/C, and it is recommended to accept this paper. However, the authors are not satisfied with the following points. Minor modifications and experiments should be performed.

1.The authors have not shown the composition ratios of the Fe3N@CNTs/CC products. The authors should show the composition by ICP analysis, indicate the amount of iron and nitrogen introduced, and discuss the amount of Fe3N species and C-N present.

2.The authors have revealed the valence of Fe species by XPS. It should be discussed about the state of each Fe species and the presence of oxides.

3.The authors compare the ORR performance of the synthesized Fe3N@CNTs/CC with that of Pt/C catalysts. However, the authors failed to indicate the amount of Fe species used in the reaction, and the data is insufficient. The authors should discuss the amount of Fe species and whether the amount of Fe species is too much or too little.

4.The authors should unify the unit notation of Figure S3 with other Figures.

5.There is a lack of uniformity in some of the notations. For example, Line 182t(707.8eV). The authors need to correct these typographical errors.

Grammar and spelling are generally fine, but the notation of  unit is not consistent and should be corrected.

Reviewer 2 Report

The article "In-Situ Growth of Fe3N Nanoparticles Encapsulated N-Doped Carbon Nanotubes on Biomass-Derived Carbon Cloth as Self-Standing Electrocatalyst for Oxygen Reduction Reaction" is enlightening on the production and study of catalytically active materials based on carbon nanotubes. The topic of obtaining active and affordable catalysts for low-temperature fuel cells based on renewable sources of raw materials is undoubtedly topical. However, several issues need to be worked out.

1) The introduction is written in insufficient detail and the number of literary sources used is small for such a dynamically developing topic. It is recommended to significantly expand the literature review indicating the novelty of the proposed solution.

2) The title of the article does not accurately reflect its content. The name indicates "In-Situ Growth", however, the authors of the study did not study the growth process of nanoparticles, especially In-Situ. In addition, the title indicates "Biomass-Derived Carbon Cloth", while this aspect is not disclosed in the article itself. It is recommended to change the title of the article.

3) The authors conduct a TEM study, but do not present a histogram of the size distribution of nanoparticles.

4) According to XRD data, it is necessary to calculate the average crystallite size using the Scherrer formula and compare it with the TEM results.

5) It is necessary to carry out an elemental analysis of the obtained sample.

6) It is necessary to describe the results of XPS in more detail, it is necessary to determine the atomic content of each element in the material, in particular nitrogen. In addition, it is necessary to determine the nitrogen content of each of the 4 different types since they have different catalytic activity. It is necessary to compare elemental analysis data and surface composition results based on XPS.

Round 2

Reviewer 2 Report

The authors corrected the article and responded to comments. The article may be accepted for publication.

A small clarification on the text on page 5, line 182

"The average crystallite size calculated by the Scherrer formula is 52.01 nm" needs to be corrected to 52 nm because this method is not as accurate.